# Terahertz-Based Method for Accurate Characterization of Early Water Absorption Properties of Epoxy Resins and Rapid Detection of Water Absorption

**DOI:** 10.3390/polym13234250

**Published:** 2021-12-03

**Authors:** Hongchuan Dong, Yunfan Liu, Yanming Cao, Juzhen Wu, Sida Zhang, Xinlong Zhang, Li Cheng

**Affiliations:** 1State Grid Economic and Technological Research Institute Co., Ltd., Beijing 102200, China; fw9866@126.com (H.D.); litao199811@outlook.com (Y.C.); 18535965807@163.com (J.W.); 2State Key Laboratory of Power Transmission Equipment & System Security and New Technology, Chongqing 400044, China; zsdstar@126.com (S.Z.); zxl18789198200@163.com (X.Z.); chengl9877@gmail.com (L.C.)

**Keywords:** epoxy resins, Langmuir, terahertz, molecular simulation, prediction

## Abstract

Moisture is detrimental to the performance of epoxy resin material for electrical equipment in long-term operation and insulation. Therefore, moisture absorption is one of the critical indicators for insulation of the material. However, some relevant test methods, e.g., the direct weighing method, are time-consuming, and it usually takes months to complete a test. For this, it is necessary to have some modification to save the test time. Firstly, the study analyzes the present prediction method (according to ISO 62:2008). Under the same accuracy, the time required is reduced from 104 days to 71 days. Subsequently, the Langmuir curve-fitting method for water absorption of epoxy resin is analyzed, and the initial values of diffusion coefficient, bonding coefficient, and de-bonding coefficient are determined based on the results of molecular simulation, relevant experiment, and literature review. With the optimized prediction model, it takes only 1.5 days (reduced by 98% as compared with the standard prediction method) to determine the moisture absorbability. Then, the factors influencing the prediction accuracy are discussed. The results have shown that the fluctuation of balance at the initial stage will affect the test precision significantly. Accordingly, this study proposes a quantitative characterization method for initial trace moisture based on the terahertz method, by which the trace moisture in epoxy resin is represented precisely through the established terahertz time-domain spectroscopy system. When this method is used to predict the moisture absorbability, the experimental time may be further shortened by 33% to 1 day. For the whole water absorption cycle curve, the error is less than 5%.

## 1. Introduction

Epoxy resin has many advantages, e.g., high insulation strength, good chemical properties, and excellent environmental adaptability [1,2,3,4,5,6,7]. It is widely used in electrical equipment as a packaging and pouring material. Existing studies have shown that moisture will accelerate the aging of epoxy resin material for power equipment and reduce its insulation performance [8,9]. After the intrusion of water molecules, physical and chemical changes such as plasticization and hydrolysis will occur in epoxy resin, resulting in irreversible crack, structural damage, and performance degradation of epoxy resin [10,11,12]. Chemical aging of epoxy resin mainly involves its changes in chemical structure, including chemical processes such as bond breaking and cross-linking degradation caused by hydrolysis. Chemical changes will have an irreversible effect on the material itself. Physical changes refer to morphological changes such as stiffness degradation and crack, as well as damage on epoxy matrix. After swelling for water absorption, the crack will be enlarged with the diffusion of water molecules, and thus the moisture absorption will be accelerated, making the water molecules fully contact with the epoxy resin and further damage its matrix [13,14,15]. Therefore, moisture absorption is an important indicator for material aging, and it is necessary to characterize the moisture absorption of epoxy resin material so that epoxy resin of excessively high water absorption will not be used for the power grid.

Presently, the main method for studying water absorption of composite materials is weighing method, which may reflect the moisture absorbability of the material simply and intuitively. However, for organic polymer materials such as epoxy resin, usually, it takes more than 50 days to reach the saturation point for water absorption, which has seriously restricted the application of water absorption method in engineering practice [6,16,17,18,19]. In order to reduce the experimental time by the direct method for determining the moisture absorption and improve the practical utility of the method. ISO 62:2008 [20] put forward a water absorption prediction formula, combined with Fick’s law and the curve method or calculation tool, that can estimate the saturated water absorption of the material. However, it is still time consuming, the error is large, and the prediction effect is not obvious. Based on Fick’s Law, Simon Heid-Jørgensen [21] established the analytical homogenization model to predict the water absorption of epoxy-glass composites, which predicted the first stage of water absorption of the material successfully. However, for the second and third stages of water absorption, the prediction effect was unsatisfactory. Fu Yingqiang [22] predicted the water absorption of modified polyvinyl alcohol with the BP neural network method. The water absorption of the material was predicted successfully based on the measured water absorption values with a precision of 85%. Hui Li [23] established a water absorption prediction model for the composite in combination with Arrhenius’s and Fick’s Laws, which accurately predicted the moisture absorption of the composite any time as soaked in water at a constant temperature of 95 °C.

The existing moisture absorption prediction methods are based solely on experimental data, which has the following shortcomings: (1) for the vast majority of composite materials, relevant theory has proved that the moisture absorption should satisfy the Langmuir model [24]. However, the existing methods fail to consider the Langmuir model, and thus the prediction efficiency is low; that is, the experimental time is not shortened obviously. (2) For a certain material, a lot of tests need to be carried out first to determine its moisture absorption at different time points, thus causing heavy workload and poor universality.

In order to reduce the time required for moisture absorption test, this study conducted an epoxy resin moisture absorption prediction and proposed a shrinkage–expansion prediction algorithm based on the Langmuir diffusion model, which was conducive to engineering moisture absorption evaluation, because it shortened the experimental time greatly as guaranteeing the precision. This study first applied the water absorption prediction model of material proposed in ISO 62:2008 to the experimental samples and carried out a relevant error analysis. Then, we proposed a nonlinear fitting method based on shrinkage–expansion theory according to the characteristics of the Langmuir diffusion equation for polymers, explored the diffusion process for moisture absorption of epoxy resin through molecular simulation, and determined the search range of shrinkage–expansion algorithm. Additionally, through a comparison with the moisture absorbability of epoxy resin material measured experimentally, the study analyzed the relationship between the experimental time and the final prediction error, put forward the shortest experimental time, and obtained a preliminary moisture absorption behavior prediction model for epoxy resin. At the same time, in order to further reduce the influence for fluctuation of balance in the experiment and represent the initial trace water absorption in epoxy resin was represented precisely. The study established a terahertz time-domain spectroscopy system. As a result, the early water absorption of epoxy resin was represented precisely by the Terahertz time-domain spectroscopy test method, a theoretical verification was carried out through molecular simulation. Therefore, the prediction model was deeply optimized. The experimental time was further reduced to 24 h by terahertz spectroscopy of epoxy resin, and the precision of the model was up to 99%.

## 2. Materials and Methods

### 2.1. Materials and Experiments

In this experiment, diglycidyl ether of bisphenol-A (DGEBA) (Changzhou Runxiang Chemical Co., Ltd., Changzhou, China) was adopted, with methylhexahydrophthalic anhydride (MeHHPA) (Shanghai Aladdin Biotechnology Co., Ltd., Shanghai, China) as the curing agent and DMP30 (tertiary amine) (Jiangsu Dandelion Biotechnology Co., Ltd., Jiangsu, China) as the accelerant. The preparation process was as shown in Figure 1. Mix all the reagents according to the ratio of epoxy resin: curing agent: toughening agent: accelerant = 100:80:10:1, and stir at a constant temperature of 50 °C. Then, pour the mixture into a mold, and after vacuum defoamation, solidify at 90 °C and 110 °C for 2 h, respectively, in a thermostat (Shanghai qixin scientific instrument co., LTD, Shanghai, China). Cool and demold until flake samples of diameter 100 mm and thickness 1 mm are obtained.

The experimental process is shown in Figure 2, in which experimental samples are placed on the hollow iron frame in the constant temperature and humidity box. Before the experiment, the epoxy resin samples were dried at 60 °C in a drying oven and weighed regularly until the sample mass did not change. The average weight of the six samples after drying was recorded as $m_{0}$. Then, the constant temperature and humidity box was tested. The temperature and humidity meter was placed in the box, and the parameters were set as temperature = 25 °C and relative humidity = 95 %. When the display for the constant temperature and humidity chamber as well as the reading of built-in hygrothermograph remained unchanged, it was determined that the experimental environment was good, and the experimental test was carried out [25]. In the process of the experiment, all the dried samples were put into the constant temperature and humidity box, keeping the relative humidity unchanged; a high-precision balance (precision 1 mg) was regularly used to measure the weight, and the samples were put back immediately after each measurement. Six samples were repeated five times, and the average *m*(*t*) of five experiments was taken; then, the water absorption percentage, $w\left(t\right)$, can be expressed by Formula (1).
(1)$$w\left(t\right)=\frac{m\left(t\right)-m_{0}}{m_{0}}\times 100\%,$$
where $m\left(t\right)$ represents the mass of sample after experimental time *t*, and $m_{0}$, the mass of dried sample.

### 2.2. Molecular Dynamics Simulation

In the early stage of moisture absorption process for the material, the water absorption rate is fast, and the initial moisture absorption curve is close to a straight line. Therefore, in this study, an epoxy resin containing water molecules/anhydride curing agent system was established, and the search range of diffusion coefficient D was determined through molecular dynamics simulation.

The system was universal DGEBA, with MeHHPA as the curing agent and DMP30 as accelerant. The model construction was as shown in Figure 3. Based on the flow for epoxy resin processing, the crosslinking temperature was determined as 580 K, the crosslinking pressure as 0.005 GPa, and the COMPASS field was selected as the crosslinking force field. The standard Forcite/COMPASS force field was for several times of epoxy resin molecule optimization. The system was performed a 500 ps of molecular dynamics equilibrium simulation, in which the equilibrium was determined based on the fluctuation ranges of temperature and energy. The epoxy resin containing water molecules/anhydride curing agent system contains a total of 2397 atoms, i.e., 274 O, 975 C, and 1148 H.

The range for initial value of diffusion coefficient D was determined by simulating the diffusion of water molecules at different temperatures in the epoxy resin system.

The chain movement of water molecules in epoxy resin may be represented by mean square displacement:(2)$$MSD=\langle {\left[r_{i}(t)-r_{0}(t)\right]}^{2}\rangle ,$$
where $\;\;r_{i}\left(t\right)\;\mathrm{and}\;r_{0}\left(t\right)$ represent the position vectors of atom *i* at time *t* and time 0, respectively.

The diffusion coefficient of water molecules in epoxy resin may be solved by the Einstein Formula as follows:(3)$$D_{a}=\frac{1}{6N_{a}}\underset{t\to \infty }{\lim}\frac{d}{dt}\sum_{i=1}^{N_{a}}\langle {\left[r_{i}(t)-r_{0}(t)\right]}^{2}\rangle ,$$
where ${\vec{r}}_{i}\left(t\right)\;\mathrm{and}\;{\vec{r}}_{0}\left(t\right)$ represent the position vectors of atom *i* at time *t* and time 0, respectively. In Formula (3), when t is large enough, the diffusion coefficient D may be calculated with the mean square displacement:(4)$$D=\frac{{\left|\vec{r}(t)-\vec{r}(0)\right|}^{2}}{6t}=\frac{a}{6}$$

In Formula (4), *a* represents the slope of the fitting curve.

As shown in Figure 3, the MSD at 358 K was at maximum, and at 298 K was at minimum.

### 2.3. Terahertz Time-Domain Spectroscopy System

Considering that the requirement for initial experimental data was high for the model, this study represented water absorbability of epoxy resin precisely by terahertz time-domain spectroscopy to reduce the influence for fluctuation of balance. Due to the strong absorption effect of water molecules on terahertz waves, the nondestructive and sensitivity of terahertz technology to water molecules have attracted much attention and research from all walks of life. In recent years, a large number of articles [26,27,28,29] have studied the quantitative identification of moisture in various media by terahertz technology. Considering the stability and accuracy of THz measurement of water, we first combined it with the prediction method and proposed a faster water detection method.

The established terahertz platform was as shown in Figure 4. First, the femtosecond laser emitted a pulse less than 80 Fs at 1560 ± 20 nm, which was split into two beams perpendicular to each other by a beam splitter: pump light and the probe light. Then, the pump light was focused on the base surface of the photoconductive antenna through the reflector and time delay device, thus generating the THz pulse, which was focused on the tested sample after the parabolic mirror collimation. The THz pulse carrying sample information was transmitted through the samples, and then collimated and focused through another pair of parabolic mirrors, passing through a detector in alignment with the probe light. At last, the detector sent the signal to a computer for further data analysis.

### 2.4. DFT Methods

In order to explore the absorption peak of water-bearing epoxy resin in the terahertz band and verify the experimental data, a spectrum vibration analysis on water-bearing epoxy resin system was carried out with the Gaussian software (Gaussian, Inc, Gaussian 09W) to identify the characteristic absorption frequency of water molecules in the epoxy resin theoretically. DGEBA molecules of epoxy resin were selected for the model, with water molecules added for simulation, as shown in Figure 5. The DFT method was used for relevant calculation and simulation. The adopted method was mainly B3LYP commutative correlation functional from generalized gradient approximation, with def2-SVP defined as the base group of calculation, and the diffusion and polarization functions were added for all atoms.

## 3. Study Methods

### 3.1. Physical Model of Water Absorption

Carter described a diffusion phenomenon by the Langmuir diffusion model [30]. Water molecules bind together temporarily (physically) or permanently (chemically) in the material, thus retarding the diffusion process and reducing the speed of water absorption, especially for a long-term water absorption process. Carter represented relevant physical and chemical interactions with bonding coefficient *α* and de-bonding coefficient *β*. In this case, *α*, *β* ≪ *π*^2^
*Ds*^−2^, and the water absorption can be expressed by the following formula:(5)$$\frac{w\left(t\right)}{w_{s}}=\left[\frac{\alpha }{\alpha +\beta }\exp(-\alpha t)\left(y\left(t\right)-1\right)+\exp(-\beta t)\left(\frac{\beta }{\alpha +\beta }-1\right)+1\right]\;y\left(t\right)=1-\frac{8}{\pi^{2}}\overset{\infty }{\underset{j=0}{\sum }}\left\{\frac{1}{{\left(2j+1\right)}^{2}}\exp(-(\frac{2j+1}{2}{)}^{2})\cdot \pi^{2}\cdot 4Dts^{-2}\right\}$$

### 3.2. Error Thershold Determination

In the water absorption test of epoxy resin sample, water loss is inevitable when the samples are removed from the constant temperature and humidity box. Therefore, the weight gain rate for water absorption fluctuates greatly. In the experiment, the applicable range for precision of model was determined through measuring the error, and thus, the error caused by the test itself was avoided.

The 2500 h of moisture absorption of epoxy resin samples were analyzed using the experimental apparatus as shown in Figure 6. For each sample, the analysis was repeated five times under the same environmental conditions, and the five sets of measurements were averaged at last, with the mean value as the baseline. As shown in Figure 6, the weight of the six samples after 2500 h of water absorption ranged from 9.692 g to 9.700 g, mean value 9.696 g, experimental error ±0.004 g, and the corresponding error of weight gain rate for water absorption about 5%.

### 3.3. ISO 62:2008 Prediction Method

According to ISO 62:2008 [20], the measured *D* and *C* when constant mass is not achieved are as shown in Formula 6.
(6)$$\sqrt{D}\approx \frac{1}{C_{s}}\times \frac{d}{0.52\pi }\times \frac{c\left(t\right)}{\sqrt{t}},$$
where $C_{S}—$ saturated water absorption; d— sample thickness; t— moisture absorption time; $C\left(t\right)—$ water absorption rate measured at  t.

As applied to the epoxy resin samples, its prediction precision was as shown in Figure 7. The prediction precision increased with the experimental days. Within the 5% range of threshold, the required experimental days was about 71 days, and for the samples, relevant fluctuation was large, and the prediction effect was not obvious.

### 3.4. Fitting Method Based on Langmuir Formula

Current studies have shown that the water absorption of organic polymer materials such as epoxy resin conforms to the Langmuir’s Law. Langmuir Formula is characterized by multiple parameters, cascade equation, high latitude, and high nonlinearity. Therefore, parameters D, α, β, and Ws were estimated by optimization fitting to determine the moisture absorbability of the material.

In this study, the shrinkage–expansion theory was applied to the optimization fitting of initial water absorption data of epoxy resin so as to establish the water absorption prediction model of epoxy resin. In the shrinkage–expansion theory, there are three stages, as follows: first, gradually reduce the step size in the initial search space; the second is to expand the step size for search; the third is to calculate the mean value and standard deviation for these degree points that meet relevant requirements, which are used to determine the step size of the next process and carry out iteration. The objective function can be searched in and out of multi-dimensional starting space, the search center and step size can be adjusted by the feedback information during the search process, and thus the optimal parameters for the given Langmuir Formula [31], i.e., D, α, β and Ws, may be approached by merely a few shrinkage-expansion cycles. This algorithm does not have to give the derivative or partial derivative of the formula, thus reducing the complexity of calculation, and therefore, this study introduced the shrinkage-expansion algorithm for relevant fitting. Since the initial values for the parameters were greater than 0, they were first all set to any value greater than 0.

As shown in Figure 8, for direct fitting based on all the 104 days of data, the direct fitting based on all 104 days of data had a good effect. However, water absorption prediction based on36 h and 108 h initial experimental data had a poor effect, and the error relative to the true value was large. In order to further explore the relationship between the time required fir experiment and the final prediction error, the repeated five times of experimental data for six samples were averaged, and the fitting test was conducted with the experimental data at 18 h, 36 h, 45 h, 89 h, 108 h, 174 h, 311 h, 430 h, 511 h, 625 h, 1406 h, 1731 h, 2122 h, and 2500 h. The fitting results of the experimental data at 104 days were taken as the benchmark for observing and analyzing the error.

As shown in Figure 9, the shrinkage-expansion algorithm was directly used for optimization fitting of the Langmuir formula. Based on less than 94 days of experimental data, the error for the six samples was greater than 5.1%, and so, the prediction effect was poor, and the fitting error was large, that is, the order of magnitude for the results obtained for the parameters differed greatly from that of the true values. The reason for this was that, for the shrinkage—expansion algorithm, when there are multiple parameters and the nonlinear relationship is complex, the efficiency for optimization search is low, especially when the initial value is not so appropriate, the search process from the initial point to the optimal point is relatively slow, while the Langmuir formula contains four unknowns, with D, α, β, Ws as regression parameters to be estimated. Ws is merely parameter solution > 0, and it is impossible to estimate its initial value. Therefore, it is necessary to narrow the shrinkage range and determine the initial values of D, α and β.

## 4. Determination of the Initial Values of the Parameters

### 4.1. Search Range of Diffusion Coefficient D

Table 1 showed the diffusion coefficient of water molecules at 298 K, 313 K, 328 K, 343 K, and 358 K, with R*^2^* as the goodness of fit, which was greater than 0.98 for all, indicating that the simulation fitting results are credible. The diffusion coefficient of water molecules in this study was close to that in relevant literature [32], which indicated that the simulation method used in this study was effective and reasonable. According to Table 1, the diffusion coefficient of water molecules in epoxy resin increased with temperature.

There is no unit for Slope a or correlation coefficient R^2^, where D is diffusion coefficient.

According to the classical diffusion theory, the migration rate as well as the diffusion coefficient of water molecules increases with temperature. Microscopically rising temperature makes the model expand and the free volume of water molecules increase. At the same time, with the increase in temperature, the kinetic energy of water molecules increases, and the binding effect of epoxy resin on water molecules decreases. Therefore, the diffusion coefficient of water molecules increases with temperature.

According to the thermodynamic formula, the thermodynamic process of gas molecular diffusion follows the Arrhenius Formula, that is, the diffusion coefficient of water molecules has an exponential relationship with temperature. A temperature fitting was carried out for the diffusion coefficient of water molecules, and the fitting results were as shown in Figure 3. As a result, the diffusion coefficient functions of water molecules at different temperatures were obtained, which proved the accuracy of molecular simulation results.

The theoretical value for the magnitude order of diffusion coefficient of water molecules in pure epoxy resin was determined as about 10^−7^ through the establishment of molecular model. However, in practice, the penetration rate of water molecules through the epoxy resin may be hindered by the internal forces of the material. For this, the initial magnitude order for the diffusion coefficient may be selected as 10^−7^~ 10^−8^ for selecting the initial value of the algorithm. However, it should be noted that this is only a range for the initial value of the algorithm; that is, it is not the range for final value due to the influence of other environmental factors.

### 4.2. Search Range of α and β

In the initial moisture absorption stage of epoxy resin for rapid water absorption, water molecules interacting intensively diffuse into the polymer, while free water quickly fills the free volume of the epoxy resin. After a certain moisture absorption time, the diffusion is saturated, and the free water and the bound water are in equilibrium. This process is time consuming, while for the molecular simulation method, the time is relatively short, with ps as the unit, and thus, it cannot be simulated by molecular simulation. Therefore, the range of initial value for α and β was determined by an experiment and relevant literature in this study.

As shown in Table 2, the magnitude order of α and β for water absorption of epoxy resin measured in this experiment was 10^−6^ ~10^−7^ s^−1^. Based on an enormous amount of literature, the magnitude order of α and β for organic polymer materials such as epoxy resin is roughly 10^−6^ ~10^−8^ s^−1^. Therefore, the initial value was set as 10^−7^ s^−1^ for α and β of epoxy resin.

## 5. Prediction Results for Water Absorption of Epoxy Resin

In the experiment, the moisture absorption of epoxy resin material was measured, the relationship between the required experimental time for prediction and the final prediction error was analyzed, and the shortest time required for experiment was proposed. Thus, the moisture absorption behavior prediction model of epoxy resin was obtained.

The 36 h and 108 h experimental results were applied through the prediction model after the initial values were substituted. The results were as shown in Figure 10 and Table 3. Obviously, the prediction curve and saturated water absorption were more precise than before optimization of the model (Figure 8).

The relationship between the required experimental time for prediction and the final prediction error was as shown in Figure 11. Based on the results for experimental error, the precision of the established water absorption prediction model of epoxy resin was further verified. Based on the precision of the testing apparatus mentioned above, the error threshold was determined as 5%, and thus, the shortest time required for experiment was 36 h. The prediction precision may be increased with experimental time. The optimized prediction model shortened the experimental time greatly as guaranteeing the precision as compared with the model before optimization and the prediction method proposed in ISO 62:2008.

However, the stability of prediction models for different samples needed to be further improved. As shown in Figure 11, the final error corresponding to the early experimental time of different samples fluctuated greatly, which was caused by the instability of balance. Therefore, it is necessary to further optimize the stability of the model.

## 6. Precise Representation of Early Water Absorbability

The influence of water content on terahertz absorption was analyzed through calculating the absorption spectra of the samples. The THz-TDS test system was used to obtain the reference and sample time domain signals, which were u_ref_(t) and u_sample_(t), respectively. The u_ref_(t) and u_sample_(t) were converted to frequency domain through Fourier transform, and thus, frequency functions E_ref_(ω) and E_sample_(ω) were obtained. According to Bill Lambert’s Law, the absorbance is directly proportional to the concentration of light absorbing material and the thickness of absorbing layer but inversely related to the transmittance. Therefore, Formula (7) may be used to calculate the absorption coefficient (α) of the samples.
(7)$$\alpha =\frac{1}{d}ln(\frac{A_{ref}}{A_{sample}}),$$
where d is the sample thickness, and *A_ref_* and *A_sample_* are the amplitudes of the reference signal and sample signal frequency functions, respectively.

After testing, analyzing, and median filtering of epoxy resin, the absorption coefficient of the samples with different water content was obtained, as shown in Figure 12. Obviously, there was an intensive absorption peak near 1.9 THz, and the peak value increased with water content.

The vibration spectra of the mixed system at 0–3 THz were as shown in Figure 13.

According to Figure 13, the vibration simulation results of water-bearing epoxy resin system showed obvious absorption peaks at 1.13, 1.51, 1.9, and 2.5 THz, and the intensity of absorption peak at 1.9 THz was the highest, which was consistent with the test results.

For this, the peak value at 1.9 THz was taken as the reference for fitting with water absorption as shown in Figure 14. For absorption spectra of epoxy resin samples with different water content, the peak height at 1.9 THz fitted well with water content, with goodness of fit greater than 0.98. Therefore, the device may be used for preliminary testing.

The error comparison of Terahertz with balance for testing was as shown in Figure 15, where there were water absorption data indirectly represented by terahertz and obtained by the balance. According to the obtained prediction curve, under an error threshold of 5%, the terahertz method may shorten the experimental time to 24 h and enhance relevant stability as compared with the testing of balance, thus further shortening the experimental time.

## 7. Conclusions

For the conventional experimental method, it takes a long time to test the water absorption of epoxy resin for electrical equipment. The core idea of this study is “to obtain long-term water absorption over time with short experimental time”. With the prediction model proposed in this study, the long-term water absorption may be predicted, and the water absorption curve may be drawn simply by substituting 24 h of initial experimental data into the model.

(1)Based on the nonlinearity and high latitude of the Langmuir diffusion equation, this study has proposed a shrinkage-expansion algorithm-based fitting method, which may realize the fitting of the water absorption process, but the algorithm is still to be improved, for the experimental time is merely reduced from 104 days to 94 days.(2)The diffusion coefficient of epoxy resin at different temperatures has been simulated and analyzed with a molecular simulation method, and the diffusion coefficient and the temperature satisfy the Arrhenius formula.(3)Based on the results of molecular simulation, experiment, and the relevant literature, the initial values of diffusion coefficient, bonding coefficient, and de-bonding coefficient have been determined, and the prediction model has been optimized. Thus, the experimental time is further reduced from 108 days to 1.5 days, and the prediction error is no greater than the experimental error of moisture absorption test (5%), and so the engineering requirements may be set generally.(4)In order to reduce the influence for fluctuation of balance and further shorten the experimental time required by the model, this study has proposed a method to improve the precision of the model based on terahertz. Under an error threshold of 5%, the terahertz method may shorten the experimental time from 36 h to 24 h.

## Figures and Tables

**Figure 1 polymers-13-04250-f001:**
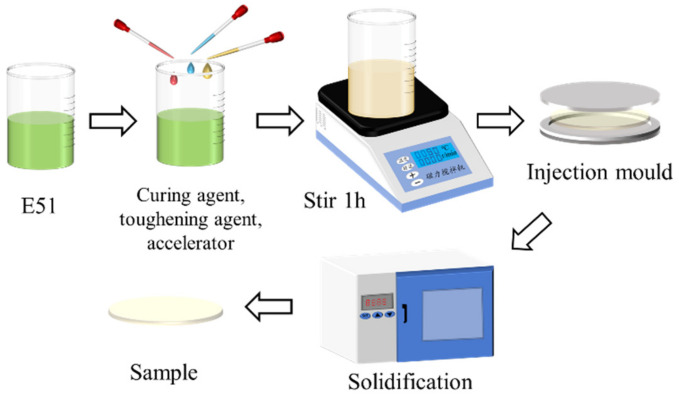
Sample preparation process.

**Figure 2 polymers-13-04250-f002:**
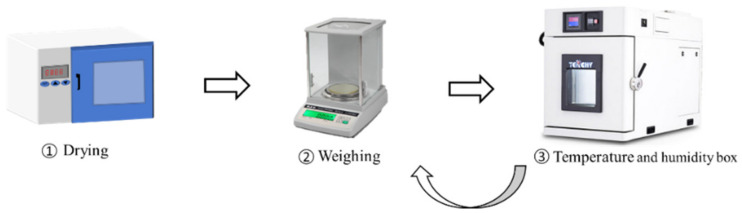
Experimental process.

**Figure 3 polymers-13-04250-f003:**
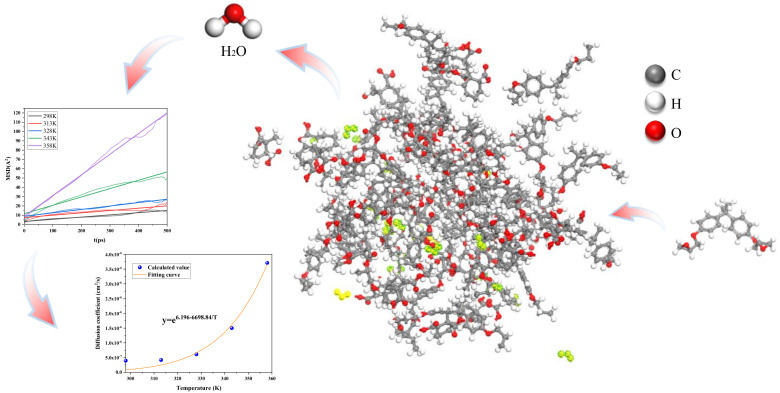
DGEBA Molecular Dynamics Model: the MSD curves at different temperatures and the relationship curves between diffusion coefficient and temperature were obtained by simulation.

**Figure 4 polymers-13-04250-f004:**
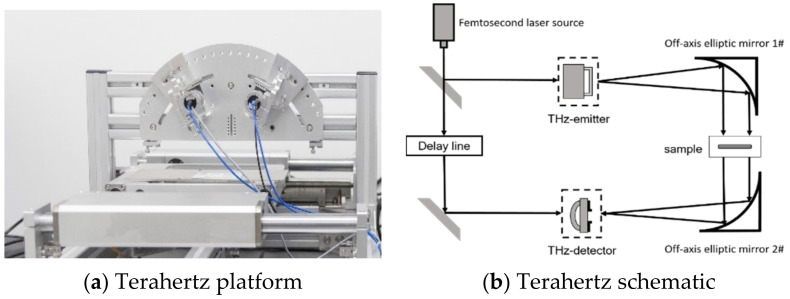
The transmission terahertz time-domain spectroscopy system.

**Figure 5 polymers-13-04250-f005:**
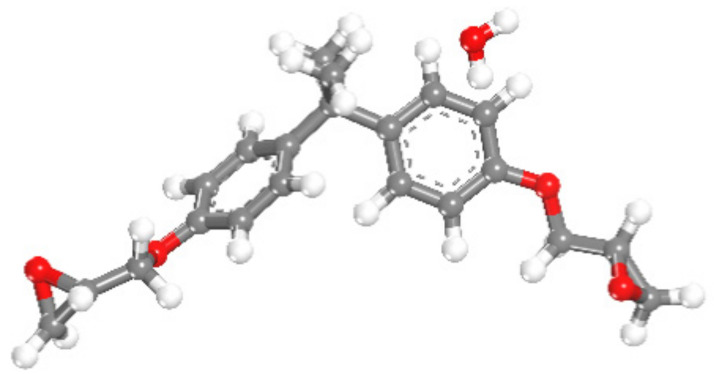
Molecular simulation of water-bearing epoxy resin system.

**Figure 6 polymers-13-04250-f006:**
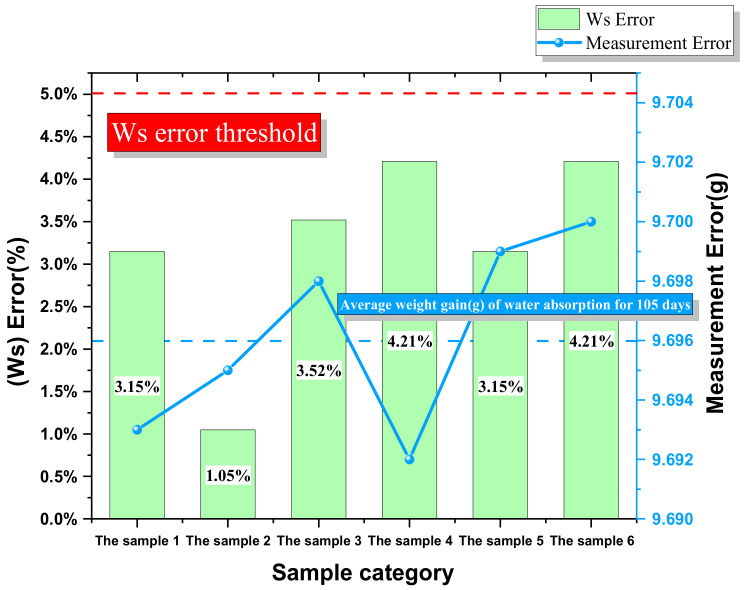
Test error of the testing apparatus. The right ordinate is the weight range of six samples after 2500 h of water absorption, and the left ordinate is the corresponding saturated water absorption error.

**Figure 7 polymers-13-04250-f007:**
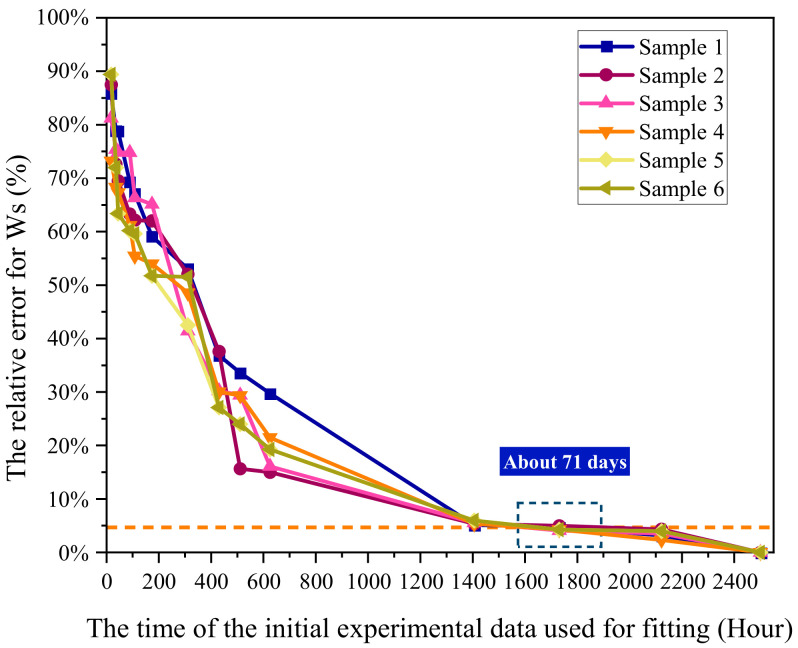
Relationship between experimental days and prediction accuracy of ISO 62:2008 model for six samples.

**Figure 8 polymers-13-04250-f008:**
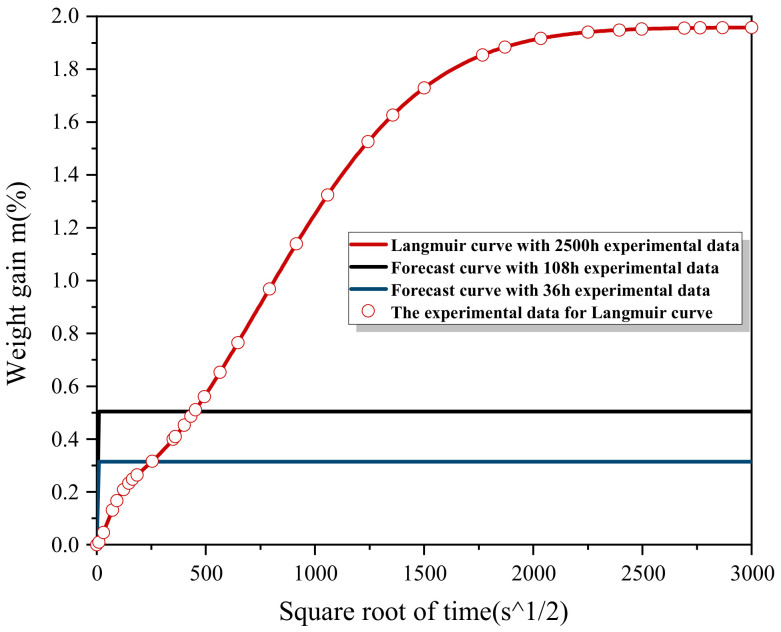
The experimental data of 36 h, 108 h, and 2500 h were used to directly apply the shrinkage-expansion algorithm to fit the obtained curves.

**Figure 9 polymers-13-04250-f009:**
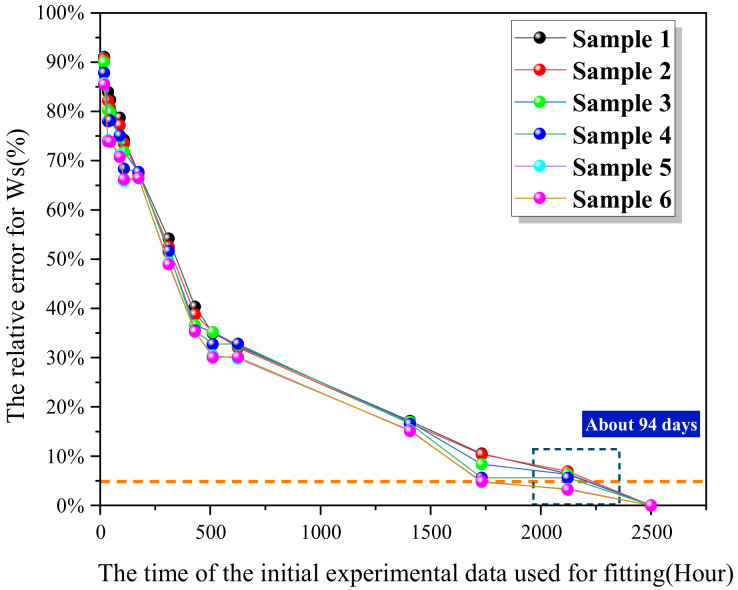
The relationship curve between experimental time and water absorption prediction error of the initial shrinkage–expansion prediction method directly applied to six samples.

**Figure 10 polymers-13-04250-f010:**
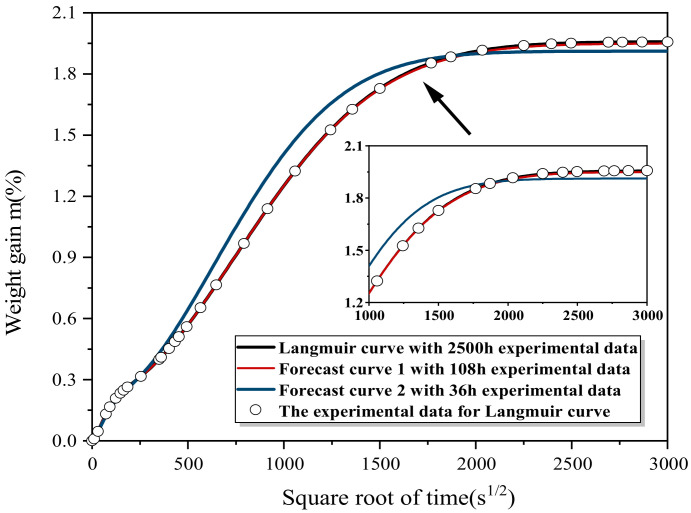
The experimental data of 36 h, 108 h, and 2500 h were used to compare the predicted curves of the optimized shrinkage-expansion algorithm.

**Figure 11 polymers-13-04250-f011:**
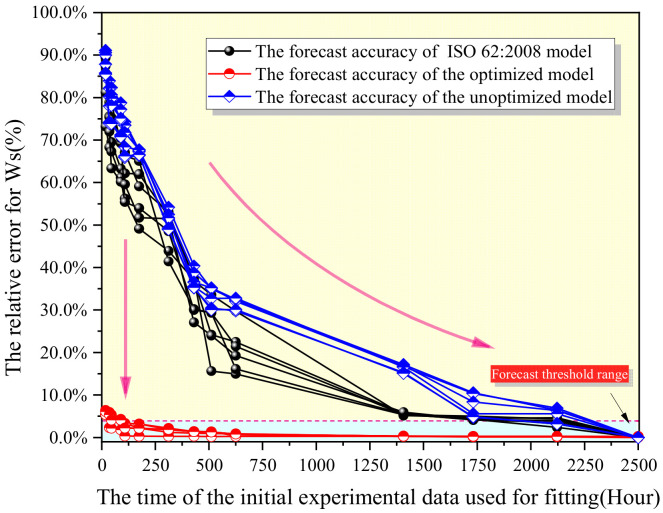
ISO 62:2008, Unoptimized Model and Optimized Model Prediction Error Comparison.

**Figure 12 polymers-13-04250-f012:**
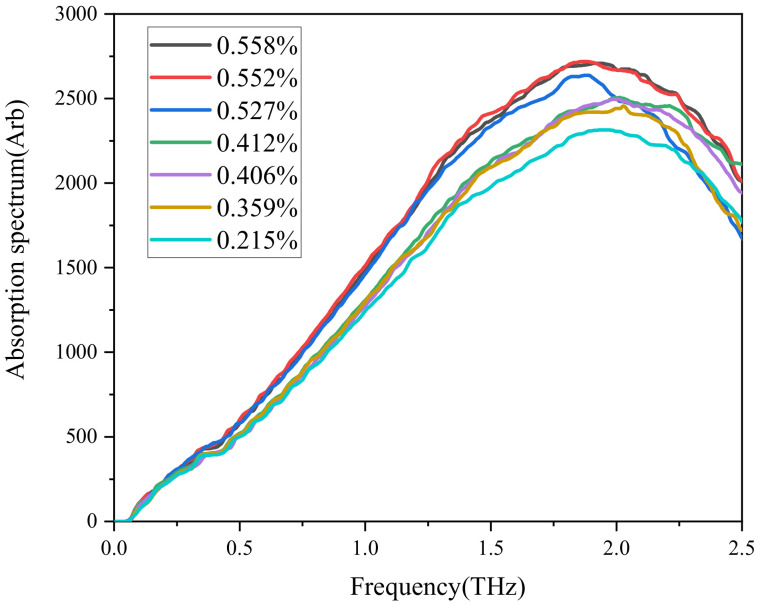
Absorption coefficient of samples with different water content.

**Figure 13 polymers-13-04250-f013:**
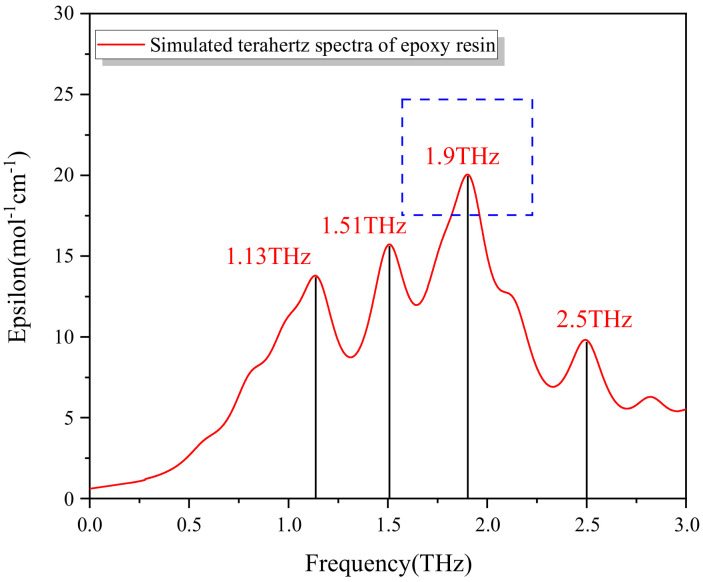
Terahertz spectrogram for molecular simulation.

**Figure 14 polymers-13-04250-f014:**
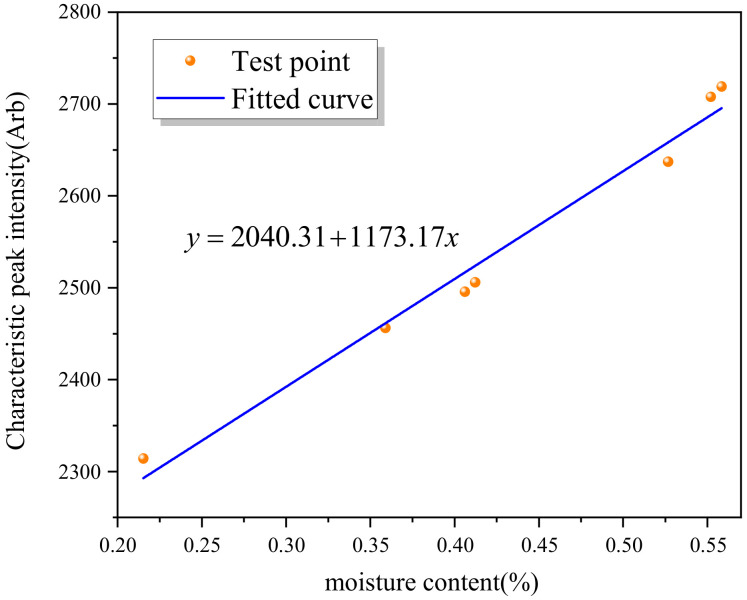
Peak height at 1.9 THz and relevant water content curve.

**Figure 15 polymers-13-04250-f015:**
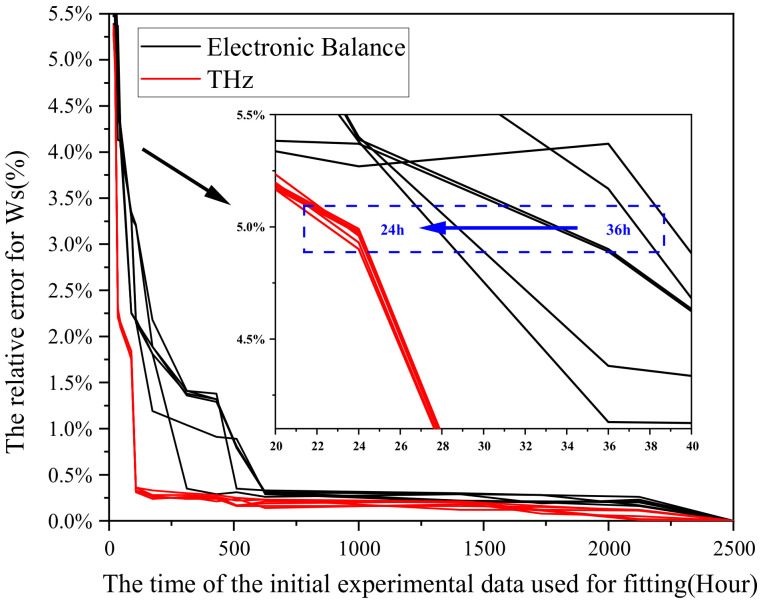
Comparison of model prediction precision between THz and electronic balance.

**Table 1 polymers-13-04250-t001:** Diffusion coefficient of water molecules at different temperatures in epoxy resin.

Temperature	a	R^2^	D/cm^2^/s
298 K	0.024	0.98	3.96 × 10^−7^
313 K	0.025	0.98	4.19 × 10^−7^
328 K	0.037	0.98	6.11 × 10^−7^
343 K	0.090	0.99	1.50 × 10^−6^
358 K	0.222	0.99	3.71 × 10^−6^

**Table 2 polymers-13-04250-t002:** Ranges of *α* and *β* of different materials.

	Water Absorption Test Value of Epoxy Resin/s^−1^	Silicone Rubber [16] /s^−1^	Epoxy Resin Adhesive (Type EC 2216 from 3M) [33]/s^−1^
α	7.432 × 10^−6^	1.5 × 10^−6^	4 × 10^−8^
β	9.071 × 10^−7^	2.5 × 10^−6^	8.1 × 10^−8^

**Table 3 polymers-13-04250-t003:** Parameter Comparison of 36 h, 108 h, and True Value.

The Experimental Time	D	The Relative Error	α/β	The Relative Error	$W_{s}$	The Relative Error
The real value	1.098 × 10^−7^	0	8.248	0	1.958	0
108 h	1.082 × 10^−7^	1.457%	8.003	2.970%	1.95	0.409%
36 h	1.052 × 10^−7^	4.189%	9.113	10.487%	1.912	2.349%

## Data Availability

Not Applicable.

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
