# Peer review of "Terahertz-Based Method for Accurate Characterization of Early Water Absorption Properties of Epoxy Resins and Rapid Detection of Water Absorption"

_polymers, 2021, doi:10.3390/polym13234250_

Round 1
Reviewer 1 Report
This is an excellent paper that deals with the rapid detection of water absorption content. The literature review was thorough, and the methods, data analysis, and findings were all good. However, I have reservations about the molecular simulation. I anticipate that because the simulation was run with a small number of atoms, we won't be able to generalized the results to verify the experimental data. This investigation also indicates a reduction in water absorption content testing time using the terahertz pump-probe technique, which they claim to be the ones who adopted for this purpose. I am not well aware if this technique is already in use in industry/laboratory for this purpose. The authors should state in their manuscript that they are the first to use this technique for this purpose. If they aren't, the significance of this work will be lessened. In any case, I approve the publishing of this work in the Journal Polymers.
Reviewer 2 Report
In this work the authors validate a new method for water absorption prediction in epoxy resins based on Langmuir curve data fitting through a shrinkage-expansion algorithm. Using this method, the water absorption can be estimated with a precision of 5% in 36 hours, instead of 108 days required when applying conventional weighing methods (ISO 62:2008). Moreover, this time can be further reduced to 24 h when data points are collected by tetrahertz time-domain spectroscopy rather than an electronic balance. The proposed method could be very useful for engineering and industrial applications.
Overall, the work is very interesting and well designed. However, the description appears confusing and the following concerns need to be addressed before I can recommend publication:
- The figure captions are merely a suggestive title and offer a poor description of the graphs. Further details should be provided for an adequate description of the presented data.
- In the discussion section there are references to the boundary conditions used in the modelling, such as DFT method, or the detailed set-up used for the spectroscopic data acquisition. I believe such technical details should be provided in the materials and methods section, while in the discussion the focus should be the data analysis and interpretation.
- What is the thickness of the samples? Did the authors investigate different thicknesses to have a more solid evidence of the validity of their method? Is there any specific range of thickness for which the model is applicable or not applicable?
- In the materials and methods section, the sample preparation description should be improved by eliminating short and disconnected sentences and harmonized with the writing style used in the manuscript.
- The English grammar and style should be thoroughly revised.
